# Performance Analysis of Dual-Hop Hybrid RF-UOWC NOMA Systems

**DOI:** 10.3390/s22124521

**Published:** 2022-06-15

**Authors:** Ahmed Samir, Mohamed Elsayed, Ahmad A. Aziz El-Banna, Imran Shafique Ansari, Khaled Rabie, Basem M. ElHalawany

**Affiliations:** 1Faculty of Engineering at Shoubra, Benha University, Cairo 13511, Egypt; ahmed.saied@feng.bu.edu.eg (A.S.); mohamed.elsayed@feng.bu.edu.eg (M.E.); ahmad.elbanna@feng.bu.edu.eg (A.A.A.E.-B.); basem.mamdoh@feng.bu.edu.eg (B.M.E.); 2James Watt School of Engineering, University of Glasgow, Glasgow G12 8QQ, UK; imran.ansari@glasgow.ac.uk; 3Department of Engineering, Manchester Metropolitan University, Manchester M1 5GF, UK; 4Department of Electrical and Electronic Engineering, University of Johannesburg, Johannesburg 2028, South Africa

**Keywords:** hybrid RF-UOWC, exponential-generalized Gamma, non-orthogonal multiple access, outage probability, optimal power allocation

## Abstract

The hybrid combination between underwater optical wireless communication (UOWC) and radio frequency (RF) is a vital demand for enabling communication through the air–water boundary. On the other hand, non-orthogonal multiple access (NOMA) is a key technology for enhancing system performance in terms of spectral efficiency. In this paper, we propose a downlink NOMA-based dual-hop hybrid RF-UOWC with decode and forward (DF) relaying. The UOWC channels are characterized by exponential-generalized Gamma (EGG) fading, while the RF channel is characterized by Rayleigh fading. Exact closed-form expressions of outage probabilities and approximated closed-form expressions of ergodic capacities are derived, for each NOMA individual user and the overall system as well, under the practical assumption of imperfect successive interference cancellation (SIC). These expressions are then verified via Monte-Carlo simulation for various underwater scenarios. To gain more insight into the system performance, we analyzed the asymptotic outage probabilities and the diversity order. Moreover, we formulated and solved a power allocation optimization problem to obtain an outage-optimal performance. For the sake of comparison and to highlight the achievable gain, the system performance is compared against a benchmark orthogonal multiple access (OMA)-based system.

## 1. Introduction

Underwater optical wireless communication (UOWC) has received substantial research interest as an efficient transmission technology for a wide range of underwater applications such as surveillance and oceanic monitoring. Many wireless data transmission techniques faced limitations while communicating underwater, including acoustic waves and radio-frequency (RF) signals. An acoustic-based underwater communication has many drawbacks such as high latency, low data rates, and high attenuation. The situation was not much different when using RF in underwater communication scenarios [1,2]. An acoustic-based underwater communication has many drawbacks such as high latency, low data rates, high bit error rates, and high attenuation. In addition, it severely suffers from malicious attacks. This is due to the fact that acoustic communication channels are uniquely designed for networks used on land; they require more sophisticated security mechanisms [3]. The situation was not much different when using RF in underwater communication scenarios [1]. The underwater RF communications suffers from high power consumption, high latency, and incompatibility between high speed and long distance. The appropriate alternative to overcome these drawbacks was to go to the use of optical waves to communicate underwater due to its advantages over its counterparts such as low latency, high data rate, and high security when operating in the wavelength range of 450 nm to 550 nm [4,5,6]. Despite these advantages, the UOWC system suffers from harsh turbulence that prompted the researchers to search for a statistical distribution model to effectively describe the underwater turbulence. In [5], a unified exponential-generalized Gamma (EGG) model that perfectly characterizes underwater channel fading was experimentally derived.

Based on the aforementioned defects resulting from the use of RF in underwater communication, the communication between the on-land and the underwater end terminals was not applicable. Therefore, the integration between RF and UOWC communication systems via relay has become indispensable [7,8,9,10,11,12]. In [7,8], the authors measured the performance of a mixed RF-UOWC transmission systems in terms of outage probability (OP), average bit error rate, and ergodic capacity (EC) for different systems models. In [9,10], the authors measured the secrecy performance of a mixed RF-UOWC system where an eavesdropper tried to intercept RF communications. The authors in [11] study the performance of a dual-hop RF-UWOC transmission system in terms of OP and bit error rate under both fixed and variable gain relaying schemes in which different detection techniques are derived. The performance analysis of a decode-and-forward (DF) based triple hop radio frequency free space optical communication-underwater optical communication (RF-FSO-UWOC) system was discussed with closed-form expressions for OP and bit-error-rate in [12].

NOMA is a spectrum access technique that has an improving impact on the spectrum efficiency of communication systems, which is considered an optimal solution for underwater internet of things (UIoT) for enabling the communication of a higher number of underwater sensors. NOMA enables simultaneous transmission of multiplexed user data using the same resources (time/frequency/code). Power domain (PD) NOMA is the most common type of NOMA, where the multiplexing is performed by assigning different power levels for the multiplexed messages based on the power allocation factor parameter at the transmitter, while the receiver needs to perform successive interference cancellation (SIC) operation to separate the messages [13,14,15]. Authors in [16,17,18] investigated the performance of NOMA assisted underwater optical communication system in terms of coverage probability and system OP. In [15], the authors considered a NOMA-based dual-hop hybrid RF-power line communication system in terms of OP and EC. Additionally, they proved the superiority of NOMA-based system over the OMA-based one.

Hybrid communication systems, where transmission propagates through different environments, are currently attracting a lot of attention. In this paper, to enhance the spectral efficiency, we propose a downlink NOMA-based dual-hop hybrid RF-UOWC system, where the source exploits NOMA to convey two messages intended for two underwater destinations in presence of imperfect SIC. To the best of our knowledge, none of the previous work in the literature has studied hybrid RF–underwater based on NOMA as a spectrum access technique. The authors in [15] have investigated the performance of a wireless/power-line communication system, while our work investigates another hybrid system where the relay works as an intermediate node between wireless and underwater mediums. There are a lot of differences between them in terms of the field of application of the two systems. Our proposed system can find applications in many underwater applications, such as offshore oil field exploration, oceanic monitoring, and data collection. On the other hand, the system in [15] may find applications in situations where the signals suffer from penetration loss within buildings and factories. In [15], the PLC link was assumed to undergo lognormal distribution with Bernoulli Gaussian noise, including both background and impulsive noise components, while this work investigated UOWC channels that are characterized by EGG fading with AWGN.

The main contributions of this paper can be summarized as follows. (1) We derived a new closed-form and asymptotic expressions for the OPs and EC, assuming that the wireless channel is characterized by Rayleigh fading with an additive white Gaussian noise (AWGN) and the UOWC links are characterized by EGG fading with AWGN. (2) We analyzed the diversity order of the OPs. (3) We proposed and solved a power allocation optimization problem to obtain an outage-optimal power allocation factor. (4) We validated the analytical derivations through Monte-Carlo simulations for varying underwater scenarios of air bubbles level (BL) under thermally uniform and temperature gradient UOWC channels, then we analyzed the impact of system parameters on the system performance. (5) Finally, we carried out a comparison between the proposed system with an OMA-based benchmark system.

The rest of the paper is organized as follows, the system model is introduced in Section 2. The performance of the considered system is analytically evaluated by deriving the OPs and ECs in Section 3 and Section 4, respectively. The proposed power allocation algorithm is provided in Section 5. Analytical and simulation results are discussed and compared with a benchmark system in Section 6. Finally, conclusions are provided in Section 7.

## 2. System Model

In this paper, we propose a downlink NOMA-based dual-hop hybrid RF-UOWC system depicted in Figure 1, where the source (*S*) is equipped with an RF interface that aims to communicate with two destinations ($D_{1}$ and $D_{2}$) equipped with UOWC interface via an intermediate decode and forward relay (*R*). The relay has an RF interface to receive from *S* and then transmit to $D_{1}$ and $D_{2}$ through the UOWC interface, where $D_{1}$ is the far or weak user and $D_{2}$ is the near or strong user. Such a scenario can find applications in many areas in the UIoT [19] (e.g., offshore oil field exploration, oceanic monitoring, and data collection). The *S*-*R* channel ($h_{w}$) is assumed to be a RF channel characterized by Rayleigh fading with AWGN and the *R*-$D_{i}$ channels ($h_{i}$) are assumed to be UOWC channels characterized by EGG fading with AWGN, where i∈{1,2}.

For the sake of improving the spectral efficiency, we assume that *S* and *R* adopt PD-NOMA for multiplexing their messages. The communication is initiated at *S* by multiplexing the two messages $x_{1}$ and $x_{2}$ intended for $D_{1}$ and $D_{2}$, respectively. The *S*-to-*R* message is $x_{S}=\sqrt{a_{1}P_{S}}x_{1}+\sqrt{a_{2}P_{S}}x_{2}$, where $P_{S}$ is the total transmitted power at *S* and $a_{i}$ is the NOMA power allocation factor for $D_{i}$ at *S*. Without loss of generality, we assume that $a_{1}>a_{2}$ and $a_{1}+a_{2}=1$. The received message at *R* through the RF link is $y_{R}=h_{w}d^{\frac{-v}{2}}x_{S}+n_{\omega }$, where the expectation of RF channels gain is $E[{\left|h_{w}\right|}^{2}]=1$, *d* is the *S*-to-*R* link distance, *v* is the RF channel path-loss exponent, and $n_{\omega }$ represents AWGN with $n_{\omega }∼\mathrm{CN}\left(0,\;\sigma_{\omega }^{2}\right)$. Utilizing NOMA concept, *R* decodes $x_{1}$ first, then applies the SIC operation, which is assumed to be imperfect, to decode $x_{2}$. So, the signal-to-interference-plus noise ratios (SINRs) for decoding $x_{1}$ and $x_{2}$ are expressed as $\gamma_{R}^{1}=\frac{a_{1}\rho_{s}d^{-v}{\left|h_{w}\right|}^{2}}{a_{2}\rho_{s}d^{-v}{\left|h_{w}\right|}^{2}+1}$ and $\gamma_{R}^{2}=\frac{a_{2}\rho_{s}d^{-v}{\left|h_{w}\right|}^{2}}{a_{1}\rho_{s}\eta d^{-v}{\left|h_{w}\right|}^{2}+1}$, respectively, where ρs=PSPSσω2σω2, and 0≤η≤1 is the residual power factor of the imperfect SIC.

In the second phase, *R* retransmits the received messages over the UOWC channels that are characterized by independent but not necessarily identical mixture EGG distribution [5]. The relay multiplexes the detected messages using PD-NOMA again, such that $x_{R}\;=\;\sqrt{b_{1}P_{R}}x_{1}+\sqrt{b_{2}P_{R}}x_{2}$, where $P_{R}$ is the total transmitted power at *R* and $b_{i}$ is the NOMA power allocation factor for $D_{i}$ at *R*. Without loss of generality, $b_{1}>b_{2}$ and $b_{1}+b_{2}=1$. The received message at $D_{1}$ through the UOWC link $h_{1}$ is $y_{D1}=\varepsilon h_{1}x_{R}+n_{u}$, where $h_{1}$ is the EEG fading of UOWC channel from *R*-to-$D_{1}$ with expectation $E[{\left|h_{1}\right|}^{2}]=1$, ε is responsivity that is considered to be unity, and $n_{u}$ represents AWGN with $n_{u}∼\mathrm{CN}\left(0,\;\sigma_{u}^{2}\right)$. Utilizing NOMA concept, $D_{1}$ decodes $x_{1}$ first. So, the SINR for decoding $x_{1}$ at $D_{1}$ is expressed as $\gamma_{D1}^{1}=\frac{b_{1}\rho_{R}{\left|h_{1}\right|}^{2}}{b_{2}\rho_{R}{\left|h_{1}\right|}^{2}+1},$ where ρR=PRPRσu2σu2.

The received message at $D_{2}$ through the UOWC link $h_{2}$ is $y_{D2}=\varepsilon h_{2}x_{R}+n_{u}$, where $h_{2}$ is the EEG fading of UOWC channel from *R*-to-$D_{2}$ with expectation $E[{\left|h_{2}\right|}^{2}]=1$. Following the NOMA principle, $D_{2}$ decodes $x_{1}$ first and then applies the SIC operation, which is assumed to be imperfect, to decode $x_{2}$. So, the SINRs for decoding $x_{1}$ and $x_{2}$ are expressed as $\gamma_{D2}^{1}=\frac{b_{1}\rho_{R}{\left|h_{2}\right|}^{2}}{b_{2}\rho_{R}{\left|h_{2}\right|}^{2}+1}$ and $\gamma_{D2}^{2}=\frac{b_{2}\rho_{R}{\left|h_{2}\right|}^{2}}{b_{1}\rho_{R}\eta {\left|h_{2}\right|}^{2}+1}$.

**Channels Distributions:** We assume that the UOWC links $h_{1}$ and $h_{2}$ are characterized by the EGG distribution [5], which models the underwater turbulence fading resulting from air bubbles and gradient of temperature in an effective manner. EGG is a weighted combination of the exponential and generalized Gamma distributions, it effectively matches the experimental results obtained under different scenarios of channel impairments of UOWC. A closed-form expression for the cumulative distribution function (CDF) of EGG distribution is given as [5]
(1)$$F_{{\left|h_{i}\right|}^{2}}\left(x\right)=w\;G_{1,2}^{1,1}\left(\frac{1}{\lambda }{\left(\frac{x}{\mu_{r}}\right)}^{\frac{1}{r}}\left|\begin{array}{c} 1\\ 1,0 \end{array}\right.\right)+\frac{1-w}{\Gamma (a)}G_{1,2}^{1,1}\left(\frac{1}{b^{c}}{\left(\frac{x}{\mu_{r}}\right)}^{\frac{c}{r}}\left|\begin{array}{c} 1\\ a,0 \end{array}\right.\right),$$
where 0<w<1 represents the mixture ratio between exponential and generalized Gamma distributions, λ is the exponential distribution scale parameter of the exponential distribution, (a,b,c) are the parameters associated with generalized Gamma distribution, and $G_{m,n}^{p,q}\left(.\right)$ is the Mejier-G function [20]. According to the receiver detection method, heterodyne detection (r=1) or intensity modulation/direct detection (IM/DD) (r=2), the electrical signal to noise ratio (SNR) is
(2)μri=Ωxir=1Ωxi2wλ2+b2(1−w)Γ(a+2c)Γ(a+2c)Γ(a)Γ(a)r=2,
where $\Omega_{xi}$ is the average SNR of the UOWC links. We assume that $\Omega_{x1}=\Omega_{x2}=\Omega_{x}$, thus $\mu_{r1}=\mu_{r2}=\mu_{r}$. The values of (w,λ,a,b,c) for different scenarios of air bubbles under thermally uniform and gradient-based UOWC channels are experimentally obtained in [5] (Table 1 and Table 2). Finally, the RF-links $h_{w}$ undergo a Rayleigh fading with AWGN noise, therefore ${\left|h_{i}\right|}^{2}$ follows an exponential distribution whose CDF is given as
(3)$$F_{{\left|h_{w}\right|}^{2}}\left(x\right)=1-e^{-x}.$$

## 3. Outage Probability Analysis

In this section, the system performance analysis in terms of OPs is presented. The OPs are defined as the probability that the received SINR falls below a certain threshold limit. We derived closed-form expressions for the outage at each destination as well as the overall system outage. Then, we derive an asymptotic expression for each of them at a high SNR regime. To gain more insight into the system performance, the outage diversity order is further derived.

### 3.1. Outage Probability $OP_{1}$

The outage event of $D_{1}$, $OP_{1}$, occurs if *R* or $D_{1}$ fails to decode $x_{1}$, which can be formulated as
(4)$$\begin{array}{c} \begin{array}{cc} OP_{1} & =1-Pr(\gamma_{R}^{1}>\gamma_{1},\gamma_{D1}^{1}>\gamma_{1})\\ \; & \overset{\left(a\right)}{=}1-Pr\left({\left|h_{w}\right|}^{2}>\frac{\tau_{1}}{\rho_{s}d^{-v}}\right)\times Pr\left({\left|h_{1}\right|}^{2}>\frac{\beta_{1}}{\rho_{R}}\right), \end{array} \end{array}$$
where (a) stems from the independence between $h_{w}$ and $h_{1}$, $\gamma_{1}=2^{R_{1}}-1$ with $R_{1}$ as the target data rate of $x_{1}$, τ1=γ1γ1(a1−a2γ1)(a1−a2γ1) under condition that $a_{1}>a_{2}\gamma_{1}$ or a1>γ1γ1(1+γ1(1+γ1), and similarly β1=γ1γ1b1−b2γ1(b1−b2γ1) under condition that $b_{1}>b_{2}\gamma_{1}$ or b1>γ1γ1(1+γ1(1+γ1). With the aid of CDFs in (Equation 1) and (Equation 3), we obtain a closed-form expression of $OP_{1}$ as in (Equation 5).
(5)$$OP_{1}=1-e^{\frac{-\tau_{1}}{\rho_{s}d^{-v}}}\left(1-wG_{1,2}^{1,1}\left(\frac{1}{\lambda }{\left(\frac{\beta_{1}}{\rho_{R}\mu_{r}}\right)}^{\frac{1}{r}}|\begin{array}{c} 1\\ 1,0 \end{array}\right)-\frac{1-w}{\Gamma (a)}G_{1,2}^{1,1}\left(\frac{1}{b^{c}}{\left(\frac{\beta_{1}}{\rho_{R}\mu_{r}}\right)}^{\frac{c}{r}}|\begin{array}{c} 1\\ a,0 \end{array}\right)\right).$$

### 3.2. Outage Probability $OP_{2}$

The outage $OP_{2}$ occurs if *R* or $D_{2}$ fails to decode $x_{1}$ or $x_{2}$; this is due to NOMA SIC concept that involves receiving $x_{1}$ and cancels it before receiving $x_{2}$. It is formulated as
(6)$$\begin{array}{c} \begin{array}{cc} OP_{2} & =1-Pr(\gamma_{R}^{1}>\gamma_{1},\gamma_{R}^{2}>\gamma_{2},\gamma_{D2}^{1}>\gamma_{1},\gamma_{D2}^{2}>\gamma_{2})\\ \; & \overset{\left(b\right)}{=}1-Pr\left({\left|h_{w}\right|}^{2}>\frac{\tau_{1}}{\rho_{s}d^{-v}},{\left|h_{w}\right|}^{2}>\frac{\tau_{2}}{\rho_{s}d^{-v}}\right)\times Pr\left({\left|h_{2}\right|}^{2}>\frac{\beta_{1}}{\rho_{R}},{\left|h_{2}\right|}^{2}>\frac{\beta_{2}}{\rho_{R}}\right)\\ \; & =1-Pr\left({\left|h_{w}\right|}^{2}>\frac{\tau }{\rho_{s}d^{-v}}\right)\times Pr\left({\left|h_{2}\right|}^{2}>\frac{\beta }{\rho_{R}}\right), \end{array} \end{array}$$
where (b) stems from the independence between $h_{w}$ and $h_{2}$, $\gamma_{2}=2^{R_{2}}-1$ with $R_{2}$ is the target data rate of $x_{2}$, τ2=γ2γ2a2−a1ηγ2(a2−a1ηγ2) under condition that $a_{2}>a_{1}\eta \gamma_{2}$ or a1<1γ1(1+γ1(1+ηγ2), similarly β2=γ2γ2b2−b1ηγ2(b2−b1ηγ2) under condition that $b_{2}>b_{1}\eta \gamma_{2}$ or b1<1γ1(1+ηγ2(1+ηγ2), $\tau =\max(\tau_{1},\tau_{2})$, and $\beta =\max(\beta_{1},\beta_{2})$. With the aid of CDFs in (Equation 1) and (Equation 3), we obtain a closed-form expression of $OP_{2}$ as in (Equation 7).
(7)$$OP_{2}=1-e^{\frac{-\tau }{\rho_{s}d^{-v}}}\left(1-wG_{1,2}^{1,1}\left(\frac{1}{\lambda }{\left(\frac{\beta }{\rho_{R}\mu_{r}}\right)}^{\frac{1}{r}}|\begin{array}{c} 1\\ 1,0 \end{array}\right)-\frac{1-w}{\Gamma (a)}G_{1,2}^{1,1}\left(\frac{1}{b^{c}}{\left(\frac{\beta }{\rho_{R}\mu_{r}}\right)}^{\frac{c}{r}}|\begin{array}{c} 1\\ a,0 \end{array}\right)\right).$$

### 3.3. System Outage Probability $OP_{sys}$

The total system outage $OP_{sys}$ occurs if *R* or $D_{2}$ fails to decode any of the two messages or $D_{1}$ fails to decode $x_{1}$. It is formulated as
(8)$$\begin{array}{c} \begin{array}{cc} OP_{sys} & =1-Pr(\gamma_{R}^{1}>\gamma_{1},\gamma_{R}^{2}>\gamma_{2},\gamma_{D2}^{1}>\gamma_{1},\gamma_{D2}^{2}>\gamma_{2},\gamma_{D1}^{1}>\gamma_{1})\\ \; & \overset{\left(c\right)}{=}1-Pr\left({\left|h_{w}\right|}^{2}>\frac{\tau_{1}}{\rho_{s}d^{-v}},{\left|h_{w}\right|}^{2}>\frac{\tau_{2}}{\rho_{s}d^{-v}}\right)\\ \; & \;\;\;\;\;\times Pr\left({\left|h_{2}\right|}^{2}>\frac{\beta_{1}}{\rho_{R}},{\left|h_{2}\right|}^{2}>\frac{\beta_{2}}{\rho_{R}}\right)\times Pr\left({\left|h_{1}\right|}^{2}>\frac{\beta_{1}}{\rho_{R}}\right)\\ \; & =1-Pr\left({\left|h_{w}\right|}^{2}>\frac{\tau }{\rho_{s}d^{-v}}\right)\times Pr\left({\left|h_{2}\right|}^{2}>\frac{\beta }{\rho_{R}}\right)\times Pr\left({\left|h_{1}\right|}^{2}>\frac{\beta_{1}}{\rho_{R}}\right), \end{array} \end{array}$$
where (c) stems from the independence between $h_{w}$, $h_{1}$, and $h_{2}$. With the aid of CDFs in (Equation 1) and (Equation 3), we obtain a closed-form expression of $OP_{2}$ as in (Equation 9).
(9)$$\begin{array}{c} \begin{array}{cc} OP_{sys}=1-e^{\frac{-\tau }{\rho_{s}d^{-v}}} & \left(1-wG_{1,2}^{1,1}\left(\frac{1}{\lambda }{\left(\frac{\beta }{\rho_{R}\mu_{r}}\right)}^{\frac{1}{r}}|\begin{array}{c} 1\\ 1,0 \end{array}\right)-\frac{1-w}{\Gamma (a)}G_{1,2}^{1,1}\left(\frac{1}{b^{c}}{\left(\frac{\beta }{\rho_{R}\mu_{r}}\right)}^{\frac{c}{r}}|\begin{array}{c} 1\\ a,0 \end{array}\right)\right)\\ \times & \left(1-wG_{1,2}^{1,1}\left(\frac{1}{\lambda }{\left(\frac{\beta_{1}}{\rho_{R}\mu_{r}}\right)}^{\frac{1}{r}}|\begin{array}{c} 1\\ 1,0 \end{array}\right)-\frac{1-w}{\Gamma (a)}G_{1,2}^{1,1}\left(\frac{1}{b^{c}}{\left(\frac{\beta_{1}}{\rho_{R}\mu_{r}}\right)}^{\frac{c}{r}}|\begin{array}{c} 1\\ a,0 \end{array}\right)\right). \end{array} \end{array}$$

### 3.4. Asymptotic Outage Probability

A deep insight on the system performance under high SNRs regime is obtained through the derivation of the asymptotic outage probabilities. A tight asymptotic expression for the CDF of the exponential and EGG distributions at high SNR are [5]
(10)$$F_{{\left|h_{w}\right|}^{2}}\left(x\right)≃x,$$
(11)$$F_{{\left|h_{i}\right|}^{2}}\left(x\right)≃\frac{w}{\lambda }{\left(\frac{x}{\mu_{r}}\right)}^{\frac{1}{r}}+\frac{1-w}{\Gamma (a+1)}{\left(\frac{x}{b^{r}\mu_{r}}\right)}^{\frac{ac}{r}}.$$

Based on (Equation 10) and (Equation 11), we derive asymptotic expressions for $OP_{1}$, $OP_{2}$, and $OP_{sys}$ as
(12)$$OP_{1}^{\infty }≃1-\left(1-\frac{\tau_{1}}{\rho_{s}d^{-v}}\right)\left(1-\frac{w}{\lambda }{\left(\frac{\beta_{1}}{\rho_{R}\mu_{r}}\right)}^{\frac{1}{r}}-\frac{1-w}{\Gamma (a+1)}{\left(\frac{\beta_{1}}{b^{r}\rho_{R}\mu_{r}}\right)}^{\frac{ac}{r}}\right),$$
(13)$$OP_{2}^{\infty }≃1-\left(1-\frac{\tau }{\rho_{s}d^{-v}}\right)\left(1-\frac{w}{\lambda }{\left(\frac{\beta }{\rho_{R}\mu_{r}}\right)}^{\frac{1}{r}}-\frac{1-w}{\Gamma (a+1)}{\left(\frac{\beta }{b^{r}\rho_{R}\mu_{r}}\right)}^{\frac{ac}{r}}\right),$$
(14)$$\begin{array}{c} \begin{array}{cc} OP_{sys}^{\infty }≃1- & \left(1-\frac{\tau }{\rho_{s}d^{-v}}\right)\left(1-\frac{w}{\lambda }{\left(\frac{\beta }{\rho_{R}\mu_{r}}\right)}^{\frac{1}{r}}-\frac{1-w}{\Gamma (a+1)}{\left(\frac{\beta }{b^{r}\rho_{R}\mu_{r}}\right)}^{\frac{ac}{r}}\right)\\ \; & \times (1-\frac{w}{\lambda }{\left(\frac{\beta_{1}}{\rho_{R}\mu_{r}}\right)}^{\frac{1}{r}}-\frac{1-w}{\Gamma (a+1)}{\left(\frac{\beta_{1}}{b^{r}\rho_{R}\mu_{r}}\right)}^{\frac{ac}{r}}). \end{array} \end{array}$$

### 3.5. Diversity Order

To gain more insight, we study the achievable diversity order (DO) of the obtained OPs. DO is the slope of ${OP}_{l}$ where l∈{1,2,sys}. According to [21], we can calculate diversity order as DOl=−limρ→∞(log(OPl)(log(OPsys∞)log(ρ)log(ρ)). It is clear from (Equation 12)–(Equation 14) that $DO_{l}\;=\;\min\left(1,\frac{1}{r}\right)$. As $\frac{ac}{r}>>1$ in all scenarios, this result is consistent with the plots in Figure 2.

## 4. Ergodic Capacity Analysis

In this section, we derive an approximated closed-form expression for the ergodic capacity (EC) of the proposed system under the condition $a_{i}=b_{i}$. The instantaneous channel capacities for the two messages, $C_{x_{l}},C_{x_{2}}$, are given by [13,22]
(15)$$\begin{array}{c} \begin{array}{cc} C_{x_{1}} & =\frac{1}{2}\log_{2}\left(1+\min\left(\gamma_{R}^{1},\gamma_{D1}^{1},\gamma_{D2}^{1}\right)\right)\\ C_{x_{2}} & =\frac{1}{2}\log_{2}\left(1+\min\left(\gamma_{R}^{2},\gamma_{D2}^{2}\right)\right). \end{array} \end{array}$$
The EC, defined as the expectation of the channel capacity, can be mathematically expressed as [21]
(16)$$EC_{x_{i}}=\frac{1}{2\;\ell n2}\int_{\gamma =0}^{\infty }\frac{1}{1+\gamma }\left[1-F_{\gamma_{j}}\left(\gamma \right)\right]d\gamma ,$$
where j∈{a,b}. The ergodic sum capacity (ESC) can be expressed as
(17)$$ESC=EC_{x_{1}}+EC_{x_{2}}.$$
In the following subsections, we derive the individual ECs.

### 4.1. Ergodic Capacity $EC_{x_{1}}$


The CDF $F_{\gamma_{a}}\left(\gamma \right)$ is given as
(18)$$\begin{array}{c} \begin{array}{cc} F_{\gamma_{a}}\left(\gamma \right) & =1-Pr(\gamma_{R}^{1}>\gamma ,\gamma_{D1}^{1}>\gamma ,\gamma_{D2}^{1}>\gamma )\\ \; & \overset{\left(d\right)}{=}1-Pr\left({\left|h_{w}\right|}^{2}>\frac{\gamma }{\rho_{s}d^{-v}\left(a_{1}-a_{2}\gamma \right)}\right)Pr({\left|h_{1}\right|}^{2}>\\ & \;\;\;\frac{\gamma }{\rho_{R}\left(a_{1}-a_{2}\gamma \right)})Pr\left({\left|h_{2}\right|}^{2}>\frac{\gamma }{\rho_{R}\left(a_{1}-a_{2}\gamma \right)}\right), \end{array} \end{array}$$
where (d) stems from the independence of the channels gain and $0<\gamma <\frac{a_{1}}{a_{2}}$. Then
(19)ECx1=12ℓn2∫γ=0a1a1a2a211+γ(1−Fhw2(γρsd−v(a1−a2γ)))×(1−Fh12(γρR(a1−a2γ)))(1−Fh22(γρR(a1−a2γ)))dγ,
then applying variable transformation of τa=γγ(a1−a2γ)(a1−a2γ) and using the exponential distribution CDF in (Equation 3) and the tight approximated EGG CDF at high SNR in (Equation 11), we can write
(20)$$\begin{array}{c} \begin{array}{c} EC_{x_{1}}=\frac{1}{2\ell n2}\int_{\gamma =0}^{\infty }\frac{e^{-\phi_{1}\tau_{a}}{\left(1-\phi_{2}{\tau_{a}}^{\frac{1}{r}}-\phi_{3}{\tau_{a}}^{\frac{ac}{r}}\right)}^{2}}{\left(1+\tau_{a}\right)\left(1+a_{2}\tau_{a}\right)}d\tau_{a}, \end{array} \end{array}$$
where $\phi_{1}=1/\rho_{s}d^{-v}$, $\phi_{2}=\left(w/\lambda \right){\left(1/\rho_{R}\mu_{r}\right)}^{\frac{1}{r}}$, and $\phi_{3}=\left(\left(1-w\right)/\Gamma \left(a+1\right)\right){\left(1/b^{r}\rho_{R}\mu_{r}\right)}^{\frac{ac}{r}}$. Using binomial expansion
(21)$$EC_{x_{1}}=\frac{1}{2\ell n2}\left(I_{1}-2\phi_{2}I_{2}-2\phi_{3}I_{3}+\phi_{2}^{2}I_{4}+2\phi_{2}\phi_{3}I_{5}+\phi_{3}^{2}I_{6}\right),$$
where
(22)$$\begin{array}{c} \begin{array}{cc} I_{K} & =\int_{\gamma =0}^{\infty }\frac{\tau_{a}^{{}^{X_{K}}}e^{-\phi_{1}\tau_{a}}d\tau_{a}}{\left(1+\tau_{a}\right)\left(1+a_{2}\tau_{a}\right)}\\ \; & =\frac{1}{a_{1}}\int_{\gamma =0}^{\infty }\frac{\tau_{a}^{{}^{X_{K}}}e^{-\phi_{1}\tau_{a}}d\tau_{a}}{\left(1+\tau_{a}\right)}-\frac{1}{a_{1}}\int_{\gamma =0}^{\infty }\frac{\tau_{a}^{{}^{X_{K}}}e^{-\phi_{1}\tau_{a}}d\tau_{a}}{\left(\left(1/a_{2}\right)+\tau_{a}\right)}, \end{array} \end{array}$$
where K∈[1,6] and $X_{K}$ is the $K^{th}$ element in the vector X=[0,(1/r),(ac/r),(2/r),((1+ac)/r),(2ac/r)]. Utilizing [23] (Equation 3.383.10), $I_{K}$ can be expressed as
(23)$$I_{K}=\frac{1}{a_{1}}\Gamma \left(X_{K}+1\right)\left[\left(e^{\phi_{1}}\Gamma \left(-X_{K},\phi_{1}\right)\right)-\left({\left(\frac{1}{a_{2}}\right)}^{x_{K}}e^{\frac{\phi_{1}}{a_{2}}}\Gamma \left(-X_{K},\frac{\phi_{1}}{a_{2}}\right)\right)\right)].$$

Substituting (Equation 23) into (Equation 21), a closed-form expression of $EC_{x_{1}}$ is obtained.

### 4.2. Ergodic Capacity $EC_{x_{2}}$


The CDF $F_{\gamma_{b}}\left(\gamma \right)$ is given as
(24)$$\begin{array}{c} \begin{array}{cc} F_{\gamma_{b}}\left(\gamma \right) & =1-pr(\gamma_{R}^{2}>\gamma ,\gamma_{D2}^{2}>\gamma )\\ \; & \overset{\left(e\right)}{=}1-\mathrm{Pr}\left({\left|h_{w}\right|}^{2}>\frac{\gamma }{\rho_{s}d^{-v}\left(a_{2}-\eta a_{1}\gamma \right)}\right)\times pr\left({\left|h_{2}\right|}^{2}>\frac{\gamma }{\rho_{R}\left(a_{2}-\eta a_{1}\gamma \right)}\right), \end{array} \end{array}$$
where (e) stems from the independence of the channels gain and $0<\gamma <\frac{a_{2}}{a_{1}\eta }$. Then
(25)ECx2=12ℓn2∫γ=0a2a2ηa1ηa111+γ(1−Fhw2(γρsd−v(a2−ηa1γ)))(1−Fh22(γρR(a2−ηa1γ)))dγ,
then applying variable transformation of τb=γγ(a2−ηa1γ)(a2−ηa1γ) and using the Rayleigh CDF (Equation 3) and the tight approximated EGG CDF at high SNR (Equation 11), we can write
(26)$$\begin{array}{c} \begin{array}{cc} EC_{x_{2}} & =\frac{1}{2\ell n2}\int_{\gamma =0}^{\infty }\frac{e^{-\phi_{1}\tau_{b}}\left(1-\phi_{2}{\tau_{b}}^{\frac{1}{r}}-\phi_{3}{\tau_{b}}^{\frac{ac}{r}}\right)}{\left(1+\eta a_{1}\tau_{b}\right)\left(1+\left(a_{2}+\eta a_{1}\right)\tau_{b}\right)}d\tau_{b}\\ \; & =\frac{1}{2\ell n2}\left(J_{1}-\phi_{2}J_{2}-\phi_{3}J_{3}\right), \end{array} \end{array}$$
where
(27)$$\begin{array}{c} \begin{array}{cc} J_{M} & =\int_{\gamma =0}^{\infty }\frac{\tau_{b}^{{}^{Y_{M}}}e^{-\phi_{1}\tau_{b}}d\tau_{b}}{\left(1+\eta a_{1}\tau_{b}\right)\left(1+\left(a_{2}+\eta a_{1}\right)\tau_{b}\right)}\\ \; & =\frac{1}{a_{2}}\int_{\gamma =0}^{\infty }\frac{\tau_{b}^{{}^{Y_{M}}}e^{-\phi_{1}\tau_{b}}d\tau_{b}}{\left(\frac{1}{\left(a_{2}+\eta a_{1}\right)}+\tau_{b}\right)}-\frac{1}{a_{2}}\int_{\gamma =0}^{\infty }\frac{\tau_{b}^{{}^{Y_{M}}}e^{-\phi_{1}\tau_{b}}d\tau_{b}}{\left(\frac{1}{\eta a_{1}}+\tau_{b}\right)}, \end{array} \end{array}$$
where M∈{1,2,3} and $Y_{M}$ is the *M*th element in the vector Y=[0,(1/r),(ac/r)]. Using [23] (Equation 3.383.10), $J_{M}$ can be expressed as
(28)$$J_{M}=\frac{\Gamma (Y_{M}+1)}{a_{2}}\left[\left({\left(\frac{1}{a_{2}+\eta a_{1}}\right)}^{Y_{M}}e^{\frac{\phi_{1}}{\left(a_{2}+\eta a_{1}\right)}}\Gamma \left(-Y_{M},\frac{\phi_{1}}{\left(a_{2}+\eta a_{1}\right)}\right)\right)-\left({\left(\frac{1}{\eta a_{1}}\right)}^{Y_{M}}e^{\frac{\phi_{1}}{\eta a_{1}}}\Gamma \left(-Y_{M},\frac{\phi_{1}}{\eta a_{1}}\right)\right)\right].$$
By Substituting (Equation 28) into (Equation 26), a closed-form expression of $EC_{x_{2}}$ is obtained.

## 5. Proposed Power Allocation Algorithm

In this section, we propose a power allocation algorithm for optimizing the system OP under the condition $a_{i}=b_{i}$, where i∈{1,2} or equivalently $\tau_{i}=\beta_{i}$ and τ=β. The proposed optimization problem is expressed as
(29a)mina1OPsys(29b)s.t.γ11+γ1<a1<11+ηγ2(29c)a1+a2=1.
We provide the following Theorem to solve Problem (Equation 29).

**Theorem** **1.**
*Problem (Equation 29) is a convex problem, and the optimal power allocation factor value is $a_{1}^{*}=\frac{\gamma_{1}\left(1+\gamma_{2}\right)}{\gamma_{1}+\gamma_{2}+\gamma_{1}\gamma_{2}\left(1+\eta \right)}$.*


**Proof.** See Appendix A. □

Figure 6 graphically verifies that the obtained result in Theorem 1 is correct. We set $R_{1}=0.5$ and $R_{2}=0.75$ as a test values, which implies that $a_{1}^{*}\approx 0.58$ mathematically, which is consistent with the optimal value in Figure 6.

## 6. Results and Discussion

In this section, we provide a detailed discussion on the derived metrics of the proposed system under varying conditions of air bubbles for both fresh/salty and thermally uniform waters under heterodyne or IM/DD detection techniques to gain more insight and highlight some conclusions. The correctness of the obtained analysis is verified via a Monte-Carlo simulation with $10^{6}$ samples. Throughout this section, we used the distribution parameters provided in Table 1 and Table 2. Unless otherwise mentioned, the system parameters are set to $a_{1}=b_{1}=0.7$, η=0.1, $R_{1}=0.5$ bits/sec/Hz, and $R_{2}=0.75$ bits/sec/Hz; d=0.8 is the normalized distance with respect to the cell radius, and v=2, $\rho_{s}=\rho_{R}=\rho$, and $\Omega_{x}=1$. In the following, we denote “Ana” as the analytical result, “Asym” as an asymptotic result, and “Sim” as Monte-Carlo simulation results.

Figure 2 presents the outage probability for the proposed system under uniform temperature salty water for both IM/DD and heterodyne techniques. As expected, it can be deduced that the OPs significantly improve when heterodyne detection is implemented compared to IM/DD. This result is due to the ability of the heterodyne receiver to overcome the UOWC link’s turbulence effects, while this leads to a more complex receiver compared to IM/DD receiver. For example, the $OP_{sys}$ of $10^{-2}$ is achieved at ρ=37 dB under the heterodyne receiver and ρ=46 dB using the IM/DD receiver. It is remarkable that the analytical and the simulation results are a match, which validates our analytical derivations. Additionally, they match the asymptotic curves at high SNR regime. In addition, to validate the DO derived in Section 3.5, we can observe that for heterodyne detection r=1, the $OP_{sys}=0.0004747$ at ρ=50 dB and $OP_{sys}=0.00004747$ at ρ=60 dB; therefore, the $OP_{sys}$ falls with a slope of log(0.0004747)−log(0.00004747)=1. Following the same procedure for IM/DD, we can observe that the $OP_{sys}=0.006119$ at ρ=50 dB while $OP_{sys}=0.001835$ at ρ=60 dB, so the $OP_{sys}$ falls with a slope of log(0.006119)−log(0.001835)≈0.5. These results are consistent with the diversity order $DO_{l}$.

Figure 3 depicts the OPs for the proposed system under uniform temperature salty water with varying air bubbles levels BL=2.4 and BL=4.7 L/min. It is clear that the increase in the level of air bubbles leads to a degradation in the OPs performance. This is due to the rise of the water turbulence. To evaluate the performance of the proposed system in this work, we compared its performance with a benchmark scheme: the OMA-based dual-hop hybrid RF-UOWC system. Figure 3 provides the comparison between the proposed NOMA-based system versus the OMA-based system under the same system settings. According to the figure, the proposed system outperforms the benchmark in terms of OPs performance. This is due to the fact that the NOMA technique is more spectral efficient than the OMA technique.

Figure 4 illustrates the influence of the residual power factor of imperfect SIC on OPs performance of the proposed system under uniform thermally salty water at BL=2.4 L/min utilizing three varying levels of η=0,0.1,0.2. We can see that the OPs performance degrades by increasing η while the best performance is achieved with the perfect SIC scenario (η=0). This is due to the fact that an increase in η leads to a higher interference level, hence the SINRs $\gamma_{R}^{2}$ and $\gamma_{D2}^{2}$ decrease while decoding the near user message. However, the SINRs $\gamma_{R}^{1}$, $\gamma_{D1}^{1}$, and $\gamma_{D2}^{1}$ are not affected by changing η.

Furthermore, Figure 5 depicts the temperature gradient (TG) and air bubbles level effect on the OPs performance. This figure investigated three different scenarios. We set BL=2.4 and TG=0.05 in case1, BL=2.4 and TG=0.15 in case2, and BL=4.7 and TG=0.1 in case3. It is clear that the higher the level of the air bubbles and/or the temperature gradient, the stronger the turbulence, leading to a OPs performance deterioration.

Figure 6 demonstrates the influence of the power allocation factor $a_{1}=b_{1}$, which varies from 0.5 to 0.99, on the OPs performance with ρ=40 dB in two varying air bubble levels of BL=2.4 and BL=4.7 L/min. We can observe that the $OP_{1}$ enhances with the increase in $a_{1}$ due to the increase of its own message power. On the other hand, the $OP_{2}$ witnesses an improvement at first with $a_{1}$ increase as $D_{2}$ needs to decode $x_{1}$ first before decoding its own message $x_{2}$. However, with the continuous increase in $a_{1}$, an inflection point is reached since increasing $a_{1}$ means decreasing the allocated power for $D_{2}$ message ($a_{2}=1-a_{1}$) that degrades the $OP_{2}$. Finally, the $OP_{sys}$ follows the same trend as $OP_{2}$ with a bit increase. Additionally, this figure graphically proves the convexity of the optimization problem in (Equation 29).

Figure 7 illustrates the influence of the residual power factor of imperfect SIC on ECs performance of the proposed system under uniform thermally salty water at BL=2.4 L/min where η=0.01, and 0.05. We can see that the $EC_{x_{2}}$ and ESC performance degrades by increasing η. This is due to the fact that an increase in η leads to a higher interference level at the decoding process of $x_{2}$. On the other hand, the $EC_{x_{1}}$ performance is not affected by changing η. The figure also shows a perfect agreement between the simulation and the obtained analytical results at high SNR with a small deviation at low SNR. This deviation is due to the usage of the tight approximated expression for the CDF of the EGG distributions at high SNR.

Figure 8 illustrates the ECs for the proposed system under uniform temperature salty water with two air bubble levels of BL=2.4 and BL=4.7 L/min. It is clear that the increase in the level of air bubbles leads to a deterioration in the ECs performance; this is due to the increase in water turbulence.

Moreover, Figure 9 shows the effect of TG on the ECs performance in salty water under the air bubbles level BL=2.4 L/min. The figure investigated two different values of TG=0.05,0.15. It is obvious that the higher the level of the temperature gradient, the stronger the turbulence, leading to a ECs performance degradation. From Figure 8 and Figure 9, we can conclude that the effect of the variation in water turbulence (BL, TG) is negligible at the high SNR regime.

Figure 10 demonstrates the influence of the power allocation factor $a_{1}$, which varies from 0.5 to 0.99, on the ECs performance to gain insight into the effectiveness and the fairness with ρ=50 dB, under uniform temperature salty water with BL=2.4. We can see that $EC_{x_{1}}$ increases as $a_{1}$ increases because the higher power allocation factor means a higher SINRs $\gamma_{R}^{1}$, $\gamma_{D1}^{1}$, and $\gamma_{D2}^{1}$, but $EC_{x_{2}}$ drops as power allocation factor increases because the SINRs $\gamma_{R}^{2}$ and $\gamma_{D2}^{2}$ degrade. Furthermore, we can see that ESC is approximately constant over the entire range of the power allocation factor, which is owing to the fact that the rate of increase in $EC_{x_{1}}$ is approximately the same as the rate of decline in $EC_{x_{2}}$.

## 7. Conclusions

In this paper, we analyzed the system performance in terms of OP and EC and optimized the OP of a downlink NOMA-based dual-hop hybrid RF-UOWC system with DF relaying under the practical assumption of imperfect SIC, where the UOWC channels are characterized by EGG distribution. We derived new analytical closed-form expressions for OPs and ECs and asymptotic expressions for the OPs and the DO. To gain more insight, we investigated the influence of system parameters on performance. Consequently, we deduced that the increase in the level of air bubbles and/or temperature gradient leads to a degradation in the OPs and ECs performances, and the outage performance improves when implementing heterodyne detection compared to IM/DD. Moreover, we investigated the feasibility of obtaining an outage-optimal power allocation factor. Finally, we carried out a comparison with a benchmark system, from which we realize that our proposed system is suitable for UIoT applications. As a future work, we may study the a multi-underwater destination system with amplify and forward relay assuming imperfect channel state information.

## Figures and Tables

**Figure 1 sensors-22-04521-f001:**
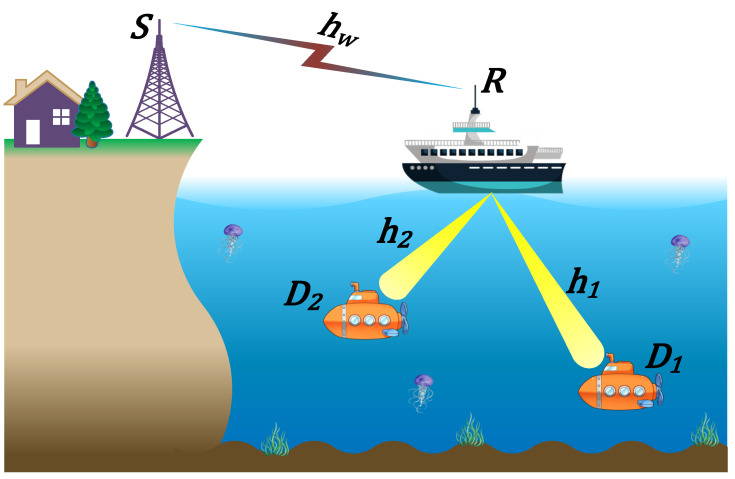
Downlink NOMA-based hybrid RF-UOWC system model.

**Figure 2 sensors-22-04521-f002:**
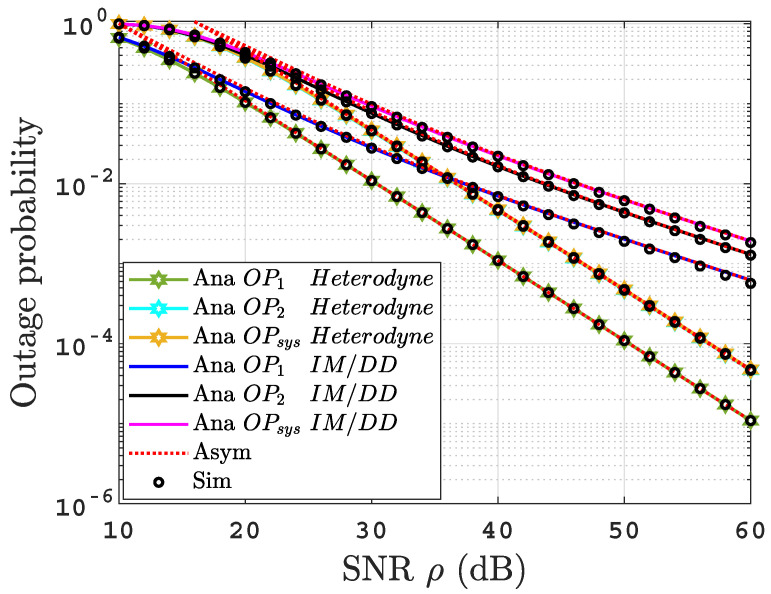
OPs versus SNR for thermally uniform UOWC links for both IM/DD as well as heterodyne detection.

**Figure 3 sensors-22-04521-f003:**
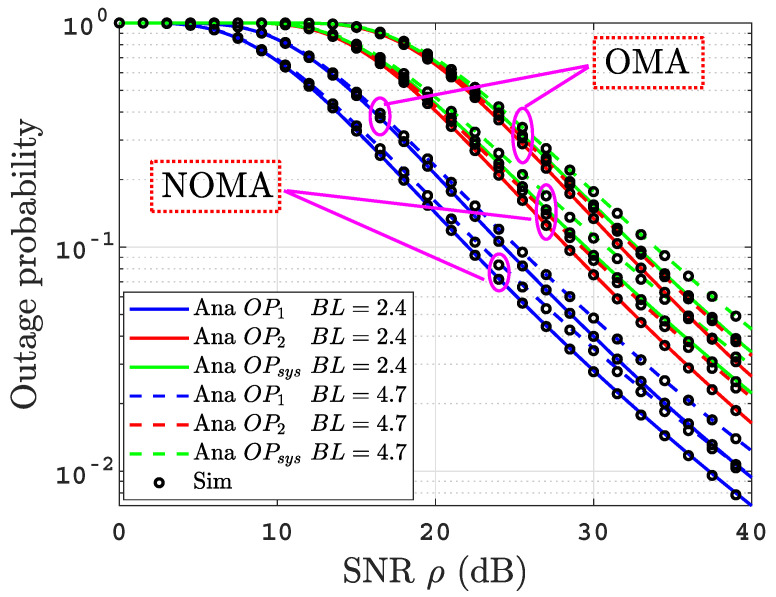
OPs versus SNR for thermally uniform UOWC links for varying air bubbles levels applicable to NOMA and OMA based systems.

**Figure 4 sensors-22-04521-f004:**
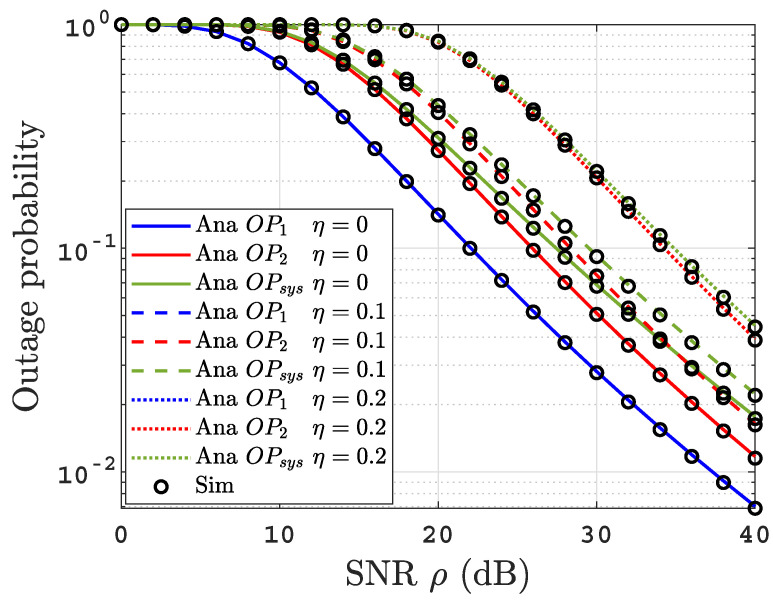
OPs versus SNR for thermally uniform salty UOWC links at *BL* = 2.4 L/min for varying values of η.

**Figure 5 sensors-22-04521-f005:**
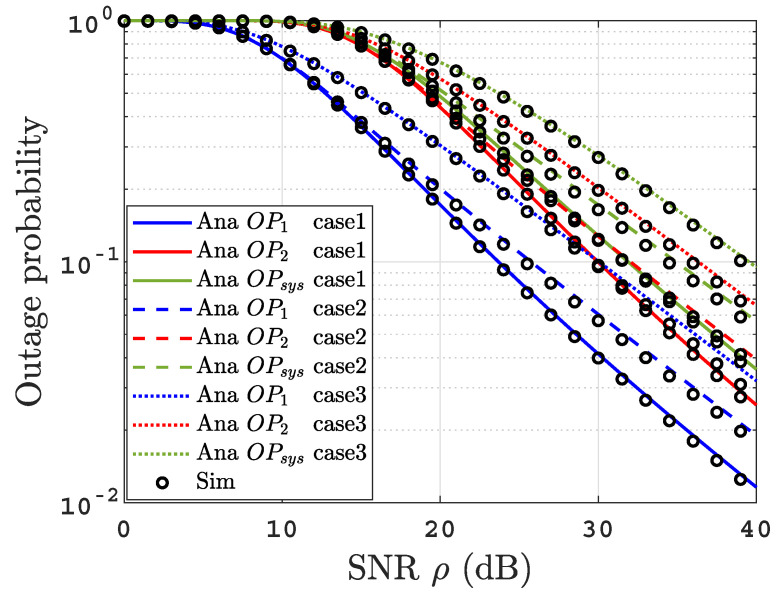
The effect of temperature gradient and air bubbles level on OPs performance.

**Figure 6 sensors-22-04521-f006:**
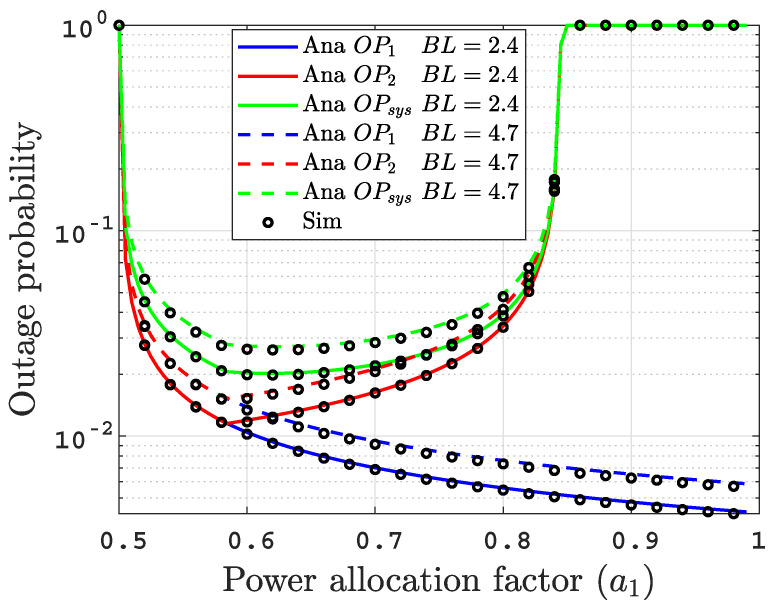
OPs over the entire range of power allocation factor at SNR = 40 dB.

**Figure 7 sensors-22-04521-f007:**
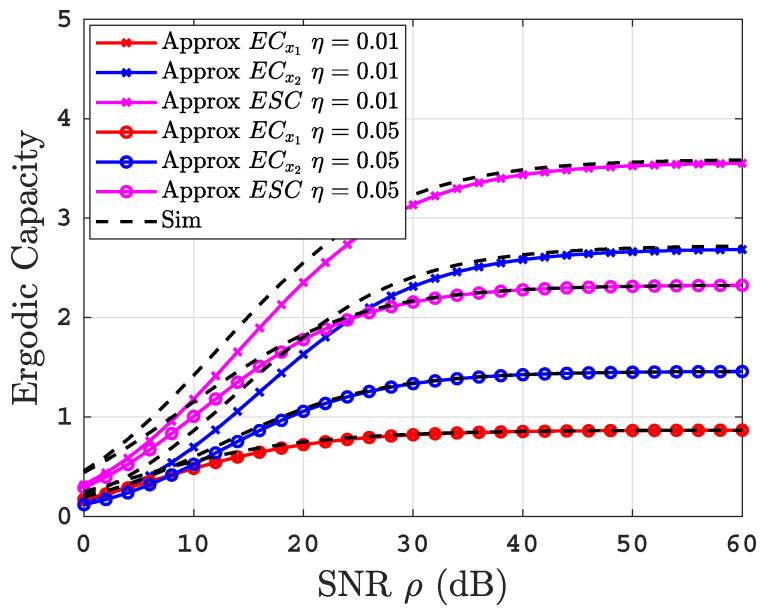
ECs versus SNR for thermally uniform salty UOWC links at *BL* = 2.4 L/min for varying values of η.

**Figure 8 sensors-22-04521-f008:**
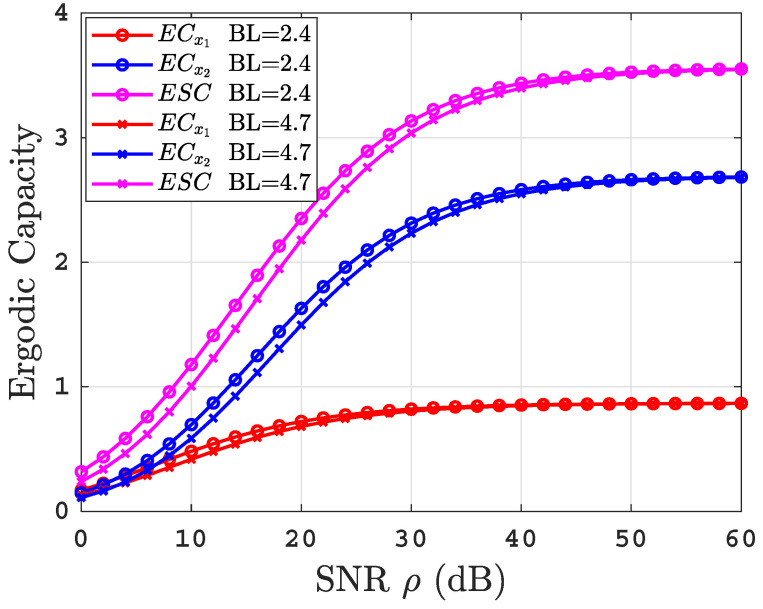
ECs versus SNR for thermally uniform UOWC links for varying air bubbles levels.

**Figure 9 sensors-22-04521-f009:**
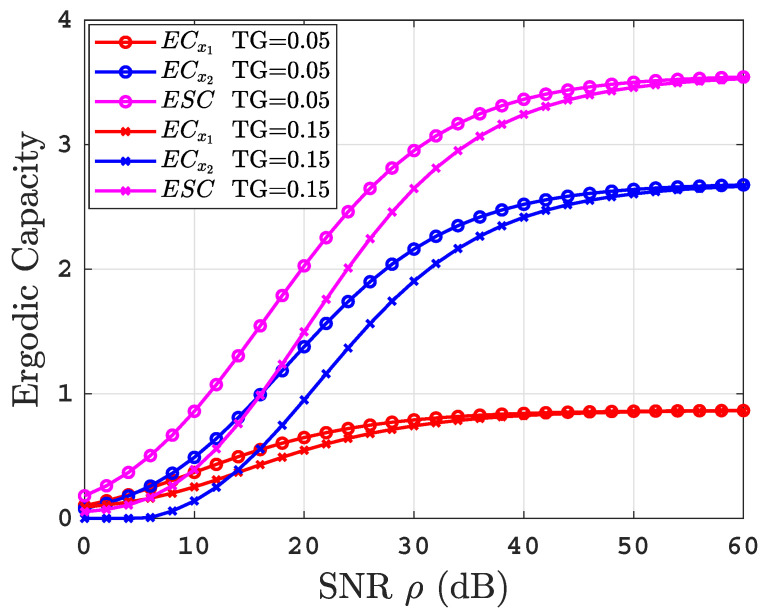
The effect of temperature gradient on ECs performance.

**Figure 10 sensors-22-04521-f010:**
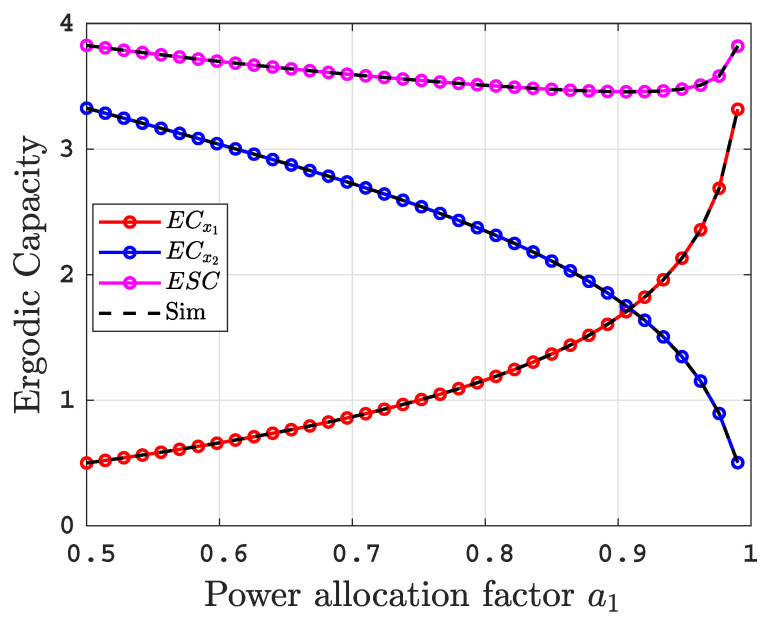
ECs over the entire range of power allocation factor at ρ=50 dB.

**Table 1 sensors-22-04521-t001:** EGG parameters for temperature gradient water [5].

*BL* (L/min)	*TG* C·cm${}^{-1}$	*w*	λ	*a*	*b*	*c*
2.4	0.05	0.2130	0.3291	1.4299	1.1817	17.1984
2.4	0.15	0.1807	0.1641	0.2334	1.4201	22.5924
4.7	0.1	0.4539	0.2744	0.3008	1.7053	54.1422

**Table 2 sensors-22-04521-t002:** EGG parameters for thermally uniform salty water [5].

*BL* (L/min)	*w*	λ	*a*	*b*	*c*
2.4	0.1770	0.4687	0.7736	1.1372	49.1773
4.7	0.2064	0.3953	0.5307	1.2154	35.7368

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
