# Peer review of "Performance Analysis of Dual-Hop Hybrid RF-UOWC NOMA Systems"

_sensors, 2022, doi:10.3390/s22124521_

Round 1

Reviewer 1 Report

1) At some places author has written [2],[3], and at some places [4]-[7], I suggest using the same format for all and using similar errors.

2) In Introduction section, the drawbacks of each conventional technique should be described clearly.

3) Introduction section can be extended to add the issues with respect to existing work.

4) How do the results of this analysis guide the design of practical communication systems? the author needs to give a brief introduction.

5) The authors have not considered the effects of distance.

6) As a conclusion, the technical content is good. Therefore, the contribution of this article is also satisfactory. I am accepting article with minor revision for publication in this journal.

Reviewer 2 Report

In this paper, the authors investigate the hybrid combination between underwater optical wireless communication (UOWC) and radio frequency (RF) as an important requirement to enable communication across the air-water boundary.

(1)The authors of this paper explore UOWC techniques and derive new closed-form and asymptotically-expressed links for OP and EC characterized by EGG fading and AWGN.

(2)The results of this paper are clearly quite consistent with the application of technology development, and the diversity order of OP contributes to the technology development of communication.

(3)This paper solves a power distribution optimization problem, and it is effective to obtain an optimal power distribution factor for power outages.

(4)The industrial application of this paper is quite shallow and worthy of research and application.

(5)There are seven keywords mentioned in this paper. In order to focus more on this article and facilitate readers’ selection, it is recommended to set no more than five keywords.

(6)In general, I think that the proposed system is consistent with the results compared with the OMA-based benchmark systems, and the manuscript can be accepted on "sensors" journal. 

Reviewer 3 Report

Considering that underwater communication is attracting great interest in the field of ocean digitization, the work presented here on the hybrid combination between optical wireless underwater communication and radio frequency based on non-orthogonal multiple access may be relevant in this field and represents an original approach. I recommend this work for publication if it contains some clarifications and improvements:

1) Line 37, use parentheses at EC.

2) All acronyms (standard and author defined) should be defined at first mention. For example, OMA is not defined in line 14 and SNR is not defined in line 115.

3) It would be nice if the authors would provide references to future work in the conclusion.

4) There is a lot of similarity between this work and the work "Performance of NOMA -based Dual-hop Hybrid Powerline-Wireless Communication Systems". The authors should cite this paper and  should explain the difference between this paper and the original published paper " Performance of NOMA -based Dual-hop Hybrid Powerline-Wireless Communication Systems". Previously published work must be listed between the references of this paper.

Round 2

Reviewer 3 Report

My principal objections to the manuscript are answered, corrected, and added as further explanations. Since I don’t have other concerns about the manuscript, the work may be accepted as it stands.